# The PEACE-Reviews dataset: Modeling Cognitive Appraisals in Emotion Text Analysis

**Gerard Christopher Yeo**
Institute of Data Science
National University of Singapore
e0545159@u.nus.edu

**Kokil Jaidka**
NUS Centre for Trusted Internet and Community
National University of Singapore
jaidka@nus.edu.sg

## Abstract

Cognitive appraisal plays a pivotal role in deciphering emotions. Recent studies have delved into its significance, yet the interplay between various forms of cognitive appraisal and specific emotions, such as joy and anger, remains an area of exploration in consumption contexts. Our research introduces the PEACE-Reviews dataset, a unique compilation of annotated autobiographical accounts where individuals detail their emotional and appraisal experiences during interactions with personally significant products or services. Focusing on the inherent variability in consumer experiences, this dataset offers an in-depth analysis of participants' psychological traits, their evaluative feedback on purchases, and the resultant emotions. Notably, the PEACE-Reviews dataset encompasses emotion, cognition, individual traits, and demographic data. We also introduce preliminary models that predict certain features based on the autobiographical narratives.[1].

## 1 Introduction

Language modeling provides insights into the linguistic features associated with how people think, feel, and behave. Some studies have modeled various psychological constructs in online text such as world beliefs (Clifton, 2020), personality (Christian et al., 2021), stress (Guntuku et al., 2019), locus of control (Jaidka et al., 2018), and even psychological disorders such as depression (De Choudhury et al., 2013). There is also work on modeling emotions using language models (Sookarah and Ramwodin, 2022). However, to understand emotional experiences of individuals comprehensively, what needs to be added in the current literature is understanding the cognitive antecedents of emotions— how people evaluate their situations that results in specific emotional experiences. These evaluations are known as cognitive appraisals (Smith and

---

[1]The dataset is accessible at https://github.com/GerardYeo/PEACE-Reviews.git

Ellsworth, 1985), and this study focuses on constructing a dataset to model such evaluations that people make about their situations.

Cognitive appraisal theories proposed that emotions are elicited from the evaluation, or appraisal, of situations along a set of motivationally-relevant dimensions (Scherer et al., 1984; Smith and Ellsworth, 1985). Consider the following scenario. Suppose Andy encountered a rude server in a restaurant that only attends politely to other customers except him. Andy might interpret this personal situation in the following way— a) the rude behavior of the server is unexpected as he expects that servers should always be polite to their customers, b) the rude server is responsible for him having such an unpleasant experience in the restaurant, and c) he finds that the rude behavior directed only to him and not to other customers is unfair. By interpreting the situation in this manner, Andy might experience an emotion like *anger*.

Specific emotions are characterized by profiles of cognitive appraisals that represent the evaluation of the person-situation relationship that is important to an individual (Lazarus et al., 1980). Emotions with a certain appraisal profile have been shown to influence decision-making and behavioral processes (Lerner et al., 2015). For example, individuals experiencing fear (an emotion that is characterized by the appraisals of high *threat* and *uncertainty*) made more risk-averse decisions about an event (Lerner and Keltner, 2001). The effect of emotions on decision-making processes and subsequent behaviors is also found in the domain of consumption (buying or using a product). For instance, emotions affect various processes such as purchase decisions (Achar et al., 2016), recommendations (Richins, 2008), and brand-switch intentions (Bui et al., 2011). Moreover, recent studies have also shown that cognitive appraisals directly and indirectly (via emotions) influence pre-purchase intentions and post-consumption behaviors (Sari, 2022;

Wang et al., 2019). Considering emotions and their antecedents (i.e., cognitive appraisals) can offer more holistic models of consumers' thoughts, emotions, and behaviors.

Online product reviews provide implicit evidence of consumer behavior. They include descriptions of subjective feelings and non-emotional factual recounts (Lee and Koo, 2012; Moore, 2015). Analyzing the emotional content of reviews can infer reviewer behaviors. However, these reviews may not reflect the necessary emotions and cognitive appraisals to study the psychological link between the two. Public datasets of product reviews do not consist of rich emotional labels (emotions and cognitive appraisals) to model the associations between consumer language and emotional constructs. Furthermore, given that most datasets comprise annotations inferred by independent annotators, it is worthwhile to pause and consider whether they offer an accurate first-hand representation of consumer appraisal. In other words, *are public datasets of product reviews suitable to study emotions, cognitive appraisals, and subsequent consumer behavior in the context of consumption?*

To address these research gaps, we conducted a pilot study of which kinds of writing prompts elicit high-quality text of emotions and cognitive appraisals and subsequently used the writing prompts derived from the pilot to collect the full dataset. Our dataset, called Perception and Experience of Appraisals and Consumption Emotions in Reviews (PEACE-Reviews). PEACE-Reviews offers review texts annotated with rich first-person emotional variables such as cognitive appraisal dimensions and emotional intensity. This work contributes to the extant literature that overlaps between the disciplines of Emotion Science and Natural Language Processing by enhancing the understanding of linguistic features of appraisals reflected in the emotional expression of autobiographical review text.

The key contributions of our work are as follows:

- A framework for evaluating appraisal and emotion elicitation through writing prompts.
- A novel dataset, the first to contain review text labeled with a large set of first-person emotional, cognitive, and trait variables, with cross-domain applications in computational linguistics, consumer psychology, computational psychology, and interaction design.
- Baseline experiments on identifying emotions

and cognitive appraisals from linguistic features using state-of-the-art language models.

## 2 Theoretical Framework

One of the primary theoretical perspectives of understanding and predicting emotions in people is Cognitive Appraisal Theory, as studied in Psychology. This theory posits that emotions are elicited from one's evaluation of situations that are of personal significance to one's well-being (Arnold, 1960; Frijda, 1987; Lazarus et al., 1980; Ortony et al., 2022; Roseman, 1984; Scherer et al., 1984; Smith and Ellsworth, 1985; Weiner, 1985). These evaluations, or appraisals, are done along a set of motivationally relevant variables. For example, if a person appraises a situation as 1) not consistent with their goals (goal conduciveness), 2) some other person is responsible for causing the situation (other-accountability), and 3) unfair (fairness), they will feel an emotion like *anger*. Therefore, instead of just knowing what emotion a person is feeling, these appraisal dimensions are essential in understanding *why* such an emotion is felt by the person. Thus, according to cognitive appraisal theories, how people appraise their situations results in specific emotions experienced. Moreover, appraisals are proposed to be central to the emotion process and are considered to be antecedents of emotions (Moors et al., 2013). Therefore, analyzing appraisals is essential in understanding what specific emotions get elicited in a situation and can also answer why a person is feeling the way he or she is feeling (Siemer and Reisenzein, 2007).

Autobiographical writing, such as reviews, reflects and communicates psychological aspects about the writer such as one's thoughts and feelings (Boyd and Pennebaker, 2015; Tausczik and Pennebaker, 2010). Words or phrases used in such writings can suggest cognitive processes (e.g. how people think about themselves or objects, their goals, and intentions), affective processes (e.g. how people are feeling), and even personality and traits (Pennebaker et al., 2003). Moreover, previous research has found that stronger emotional responses experienced in such consumption situations often lead to the posting of product/service reviews (Derbaix and Vanhamme, 2003; Hennig-Thurau et al., 2004). Thus, reviews are typically informative enough to reflect features of emotions and appraisals that underlie the text and therefore provide a suitable context for modeling consumer behavior.

In summary, motivated by how emotions and cognitive appraisals are critical in understanding consumer behavior, our broad aim is to analyze these emotional constructs in review text. The following research questions guided our dataset curation:

- How can we elicit high-quality text consisting of emotions and cognitive appraisals of consumption experiences using writing prompts?
- How do these compare to the product reviews posted to online marketplaces, such as Amazon?

## 3   Related Work

Numerous publicly available text datasets are annotated with emotion labels that follow various theoretical emotion taxonomies (Murthy and Kumar, 2021). These datasets span a wide range of domains such as tweets (Mohammad and Bravo-Marquez, 2017; Mohammad et al., 2018; Oberländer and Klinger, 2018; Wang et al., 2016), stories (Alm, 2008; Alm et al., 2005), news headlines (Buechel and Hahn, 2017; Rosenthal et al., 2019; Strapparava and Mihalcea, 2007), and personal description of emotional events (Scherer and Wallbott, 1994).

Although there are some annotated emotional textual datasets with third-person appraisal dimensions (Hofmann et al., 2020; Skerry and Saxe, 2015; Troiano et al., 2019, 2022), these are perceived appraisal ratings by third-person readers rather than first-person appraisals that are personally evaluated by the writer who experienced the situation. Third-person ratings are conceptually different from first-person ratings as they measure emotional *perception* rather than *experience*. The former ratings are indications of how a third-person (i.e. a reader) interprets a situation that someone else (i.e. a writer) is experiencing, and then from these interpretations, reason how the first-person writer is feeling (Ong et al., 2015; Skerry and Saxe, 2015). These third-person appraisal ratings might differ from how the writers personally appraise the situation in terms of the kinds of appraisal dimensions used and the intensity of the appraisal ratings. Thus, the existing datasets cannot address questions relating to first-person emotional experiences.

Datasets of products/service reviews are primarily investigated in sentiment analysis where the outcome labels are either 1) positive, negative, or neutral, or 2) an integer rating scale of 1 to 5. The Amazon product reviews dataset (Blitzer et al., 2007; He and McAuley, 2016), and the Yelp restaurant review dataset (Asghar, 2016) are widely utilized in studies that analyzed sentiments of product/services. Other review datasets include restaurant (Gojali and Khodra, 2016; Li et al., 2021; McAuley and Leskovec, 2013; Pontiki et al., 2015, 2016; Zahoor et al., 2020), and hotel reviews (Calheiros et al., 2017; Yordanova and Kabakchieva, 2017). Therefore, the lack of a product/service review dataset containing emotionally rich variables such as cognitive appraisals and emotional intensity is a primary motivation for the proposed dataset. Such a dataset can be useful not only for researchers from computational linguistics and consumer psychology to model linguistic correlates of emotional constructs and behaviors in the domain of consumption but also for practical applications in the fields of marketing, advertising, and business analytics to understand how specific product/service designs can be improved to result in targeted emotions.

## 4   Research design

This study aimed to curate a dataset of text responses labeled with rich emotional variables such as appraisal and emotional intensity. Therefore, we conducted a pilot study to evaluate which kinds of writing prompts elicit high-quality text of emotions and cognitive appraisals. In an online panel, qualified participants were assigned to one of four conditions eliciting different kinds of detail about their cognitive appraisal of a product they purchased. After evaluating the data collected for its emotional and cognitive richness through automatic content analysis methods, we subsequently used the most effective writing prompts to collect the full dataset.

## 5   Pilot Study

The purpose of the pilot study was to evaluate the quality of emotional text responses elicited by different writing prompts. Its protocol design is illustrated in Figure 1. Therefore, we employed a 2 x 2 factorial between-subjects design to examine whether a questionnaire format might yield more detailed emotional responses or whether a review format is as emotionally detailed as the questionnaire format. The two factors that are manipulated are the presence of emotion prompts (emotion prompts or no emotion prompts), and 2) response format (review or questionnaire). This design was motivated by the concern that the review format

responses might emphasize product-specific details instead of emotional reactions even when we ask participants to write about the emotional aspect of it. Another primary concern is that the review format responses might not contain text pertaining to appraisal dimensions. Therefore, the four conditions are:

- **Emotion + Review:** Presence of emotion prompts, and review format
- **Emotion + Questionnaire:** Presence of emotion prompts, and questionnaire format
- **Review:** Absence of emotion prompts, and review format
- **Questionnaire:** Absence of emotion prompts, and question format

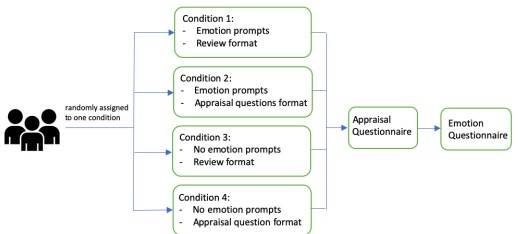

Figure 1: Schematic diagram of procedure used in the Pilot study.

The writing prompts for the different conditions are found in Table 5 of Appendix A. The context examined in the pilot and the main study is specific to using an expensive product/service. This class of products has the highest likelihood of experiencing an emotion and also results in higher emotional intensity while using it as its usage represents a situation that the participants think deeply about before purchasing and is personally important to them, compared to the experience of using a mundane or economical product such as a pen or a toothbrush (Richins, 2008). Therefore, we expect that the responses to using these products are emotionally rich.

**In the Emotion treatment,** participants were tasked to recall when they felt either the emotion of Joy or Anger when they used or experienced an expensive product/service they purchased. These emotions are chosen based on the literature on consumption emotions commonly experienced during the usage of products (Richins, 2008). Therefore, the 'presence of emotion prompts' condition asks the participants to think about an experience that involves a particular emotion.

**In the no-Emotion treatment,** participants were tasked to think about a time that they have used or experienced an expensive product/service that they purchased without any emotional prompts.

Next, we manipulate how participants are required to express themselves. **In the Questionnaire treatment,** participants will answer a series of open-ended questions about their experience with the product that they have thought about. We only included four specific open-ended appraisal questions (goal relevance, goal conduciveness, expectedness, and pleasantness) due to the feasibility of data collection (i.e. eliciting responses for a large number of appraisals might result in poor response due to fatigue) although we think that other appraisal dimensions might be relevant in the context of consumption (see Table 5 of Appendix A for the appraisal questions). The choice of these appraisal dimensions to elicit text responses is supported by previous studies that show certain appraisals that are particularly relevant to the context of consumption (Demir et al., 2009; Desmet, 2008). The design of the questions was adapted from methodological frameworks of previous studies that investigate appraisals and emotions in Psychology (Scherer, 1997; Smith and Ellsworth, 1985; Tong, 2015). We tried our best to facilitate recall by including additional prompts such as 'How did you feel when you used the product/service?' and 'What about the product that caused you to feel the emotion stated?' which might also elicit other appraisals other than the responses to the four appraisal prompts.

Finally, **in the Review treatment,** participants were tasked to write a review about their experience of using the product that they have thought about as if they were going to publish online.

After completing one of the four conditions, participants will also complete a validated cognitive appraisal and emotion questionnaire that was adapted from previous studies (Scherer and Wallbott, 1994; Smith and Ellsworth, 1985; Tong and Jia, 2017; Yeo and Ong, 2023), where they will be asked to rate how they appraised and felt when describing the experiences on a set of 20 appraisal (see Table 6 of Appendix A), and 8 emotion dimensions (see Table 7 of Appendix A), on a 7-point Likert scale. The emotions are chosen based on the literature on consumption emotions commonly experienced during the usage of products (Richins, 2008). This is to account for the fact that participants might feel multiple emotions during a con-

sumption episode.

We created a Qualtrics survey that reflects these conditions with the respective questions and prompts. Once participants have consented to participate, they will be randomly assigned to one of the conditions.

## 5.1 Data Collection

We recruited participants using the crowdsourcing called Prolific. This platform was specifically designed for academic research and has a pool of participants from the Organization for Economic Co-operation and Development (OECD) countries. Although Prolific has a pool of participants from different countries, we only recruited participants from the United States. This is to reduce the heterogeneity of the data for subsequent analyses.

A total of 18 participants were recruited in December 2022. We also set a quota to obtain an equal proportion of gender in our sample (50% male, 50% female). The pilot dataset is published on Zenodo at https://zenodo.org/records/7528896.

## 5.2 Expressed text in different conditions

One of the primary motivations for this study is to collect review text responses that express appraisal dimensions. In terms of the response format (review vs. questionnaire format), the questionnaire format generally resulted in a more detailed explanation of the participants' emotional situation (see Table 8 of Appendix A for sample responses). The average word count for texts in the questionnaire and review format is 150 and 62 words, respectively. Moreover, the participant answering in the questionnaire format provides detailed explanations on specific appraisal questions asked (e.g. why is the product consistent with what you wanted?). Thus, the text responses from the questionnaire format conditions clearly express the appraisal dimensions.

Moreover, we coded for several appraisal dimensions using the Linguistic Inquiry and Word Count 2022 dictionaries (Boyd et al., 2022) (Figure 2). LIWC22 dictionaries measure the evidence of emotional, cognitive, and social processes exemplified in writing by looking at simple word frequency counts. Their association with psychological processes, emotions, and cognitive appraisal has been well-established in prior work (Jaidka et al., 2020). Although LIWC22 does not code for cognitive appraisals per se, certain dimensions of LIWC22 might reflect cognitive appraisals. For

example, the *social* and *causation* dimensions in LIWC-22 might reflect accountability- and control-related appraisals. In general, the responses from the questionnaire format have a greater number of appraisal-related words than the review format with the exception of *fuilfil*, *need*, *social*, and *discrepancy*. Despite this, the correlations between these LIWC22 dimensions and related appraisal dimensions (Figure 5 of Appendix A) are greater for the responses in the questionnaire format. For example, the correlation between goal relevance and the needs dimension of LIWC22 for the questionnaire format is -.55 which is greater than -.12 for the review format. Therefore, appraisal ratings are generally consistent with the expressed text for the questionnaire format compared to the review format.

On the other hand, the responses received in the review format are mixed (see Table 9 of Appendix A for sample responses). Although some were very detailed and included emotional statements and appraisals, some responses were relatively short. On top of that, responses in this format typically talk about the characteristics of the products instead of the emotional experience of using the product.

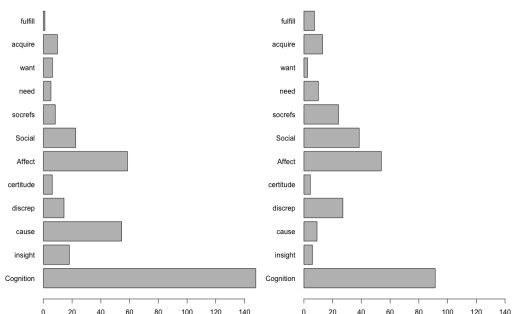

Figure 2: Means of selected LIWC-22 dimensions comparing the questionnaire and review formats, respectively. "socref" refers to "social referents," "discrep" refers to "discrepancy." The full description of the categories, their definitions, and contents are available in the LIWC 2022 manual.[2]

## 5.3 Linguistic feature comparison

Figure 3 compares the presence of words denoting cognitive processes in the pilot study to topically similar reviews in the Kaggle Amazon dataset (N = 18078), matched through a keyword search of the products mentioned in the pilot study. The first barplot compares the distribution of Affect scores, which comprises words used for both positive and

negative tones and emotional expressions, swear words (*good, well, love, happy, bad, wrong, hate*), and Cognition scores, which denote styles of thinking and processing information (*know, how, think, feel*), discrepancies (*would, can, want, could*), comparison (*but, not, if, or*), and so on. The second barplot compares the differences in the psychological states reflected in the text. Psychological states indicate signals of acquisition (*get, take, getting*), fulfillment (*enough, full, complete, extra*) and wanting (*want, hope*).

The barplots show how the text elicited through the questionnaire format overshadows Amazon reviews in terms of the simple average proportion of words that reflect emotional and cognitive appraisal. A two-tailed Wilcoxon signed-rank test of independent samples found that the level of cognition words reported in the two questionnaire conditions was significantly higher than the corresponding number for Amazon reviews ($p < 0.001$). Moreover, across all dimensions of psychological states, the questionnaire format has a greater average proportion of words compared to Amazon reviews except for the dimension of *acquire*.

## 6 Main Study

Based on the results of the pilot study, we observed that indeed the structured nature of the questionnaire format elicits richer emotional responses that contain appraisal dimensions that could be used to analyze such text and so we chose the questionnaire format as the writing prompts to elicit appraisal and emotional text responses in the main study. In terms of the presence of emotion prompts, although there were generally no significant differences between questionnaire conditions with or without emotion prompts (with the exception of the *cognition* dimension of LIWC22), we decided to go along with the presence of emotion prompts as it allows us to determine the ground truth emotion labels for each participant's experiences.

Prior to the main data collection, we conducted a screening study that asked one question-whether participants have purchased expensive products/services recently. Participants who answered yes are then eligible to participate in our main study. More broadly, this question serves as a proxy for products/services that are personally important to the participants, which we are interested in as this class of products has the highest likelihood of experiencing an emotion while using

it (Richins, 2008).

During the main data collection, participants followed the framework of the presence of emotion prompts and questionnaire format. That is, participants were tasked to state the dominant emotion experienced when they were using the expensive product/service. Unlike the pilot study where the emotion prompts involved only the emotion of joy or anger, participants were instead asked to state the dominant emotion (amongst the eight emotions listed and an 'other' option). Following this, they answer the same set of questions as in the pilot study in the text regarding their experiences and appraisals with respect to the dominant emotion that they have stated.

Additionally, we also added a trait measure called Preference for Intuition and Deliberation Scale (Betsch, 2004) to measure individual differences in the tendency to utilize a particular decision mode. This trait measure is highly relevant in the context of consumer behavior to understand the decision-making process underlying the consumers' experiences. Finally, they answered the same validated 8-emotions and 20-appraisals questionnaires in the pilot study. We also collected other responses such as the product/service that the participants have recalled and the associated cost, intention to buy the product again, and intention to recommend the product to others.

This process was then repeated again and participants were tasked to recall another experience of using an expensive product/service that is opposite in terms of valence (positive or negative) to the experience that they have previously stated. That is, if the participants recalled a positive experience previously, they were now tasked to recall and write about a negative experience. This is typically done in appraisal studies and allows us to obtain another set of responses (Tong, 2015; Smith and Ellsworth, 1985).

### 6.1 Data Collection

A total of 700 participants were recruited in March 2023 (See Table 10 of Appendix B for demographic statistics). Since each participant provided text responses and ratings for two experiences (one positive and one negative), our dataset consists of 1,400 instances. The data can be found in the following repository- https://github.com/GerardYeo/PEACE-Reviews.git.

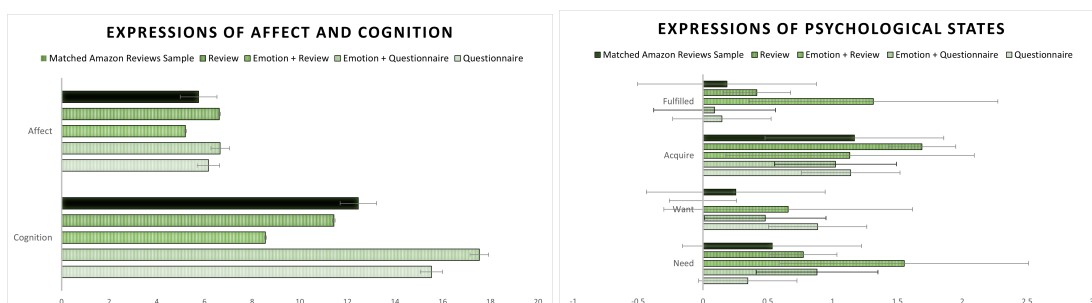

Figure 3: Distribution of linguistic features compared to an Amazon Reviews sample (N = 18,078) matched to the topics of the Pilot dataset through a keyword search.

## 6.2 Data Characteristics

### 6.2.1 Statistics

The statistics of the text responses for each prompt are presented in Table 1. When combining the text responses from all the prompts, the average number of tokens is significantly longer than previous autobiographical emotion dataset (ISEAR; (Scherer and Wallbott, 1994), and other reviews datasets (Maas et al., 2011). Datasets with a small number of tokens for each instance hinder the identification of linguistic features of a large number of appraisal and emotion constructs. On the other hand, the relatively large number of tokens for each instance in our dataset has the potential to study and segment linguistic features of appraisals and emotions in first-person emotionally rich autobiographical writings.

### 6.2.2 Analysis of appraisals and emotions ratings

In terms of ratings, we focus on appraisals and emotions, although we collected other data such as decision-making and person-level traits. For the cognitive appraisal questionnaire, there is an option for participants to indicate whether an appraisal is relevant during their experience of using the product/service. It is important to study these NA responses as not all appraisal dimensions might be relevant to the context of consumption. Table 6 shows the percentage of NA responses for each appraisal dimension. The mean percentage of NA responses across all the appraisals is 9.49%. The appraisals of goal conduciveness and pleasantness have the lowest number of NA responses which suggests that they are highly relevant in emotional experiences. On the contrary, circumstances-related appraisals might not be that relevant in the context of consumption experiences.

Table 2 indicates the distribution of the dominant emotion felt when participants were using the re-

| Prompts | Mean | Min | Max |
|---|---|---|---|
| What about the product/service that made you feel the emotion that you have stated? | 53.5 | 9 | 780 |
| Why were you feeling that emotion when you were using or experiencing the product/service? | 29.7 | 7 | 360 |
| Was the product/service important to you? Please also state the reason(s) of why was the product/service important/unimportant to you. | 27.5 | 9 | 256 |
| Was the product/service consistent with what you wanted? Please also state the reason(s) of why was the product/service consistent with what you wanted. | 25.7 | 3 | 243 |
| Was it a pleasant experience when you were using the product/service? Please also state the reason(s) of why was it a pleasant experience when you were using the product/service. | 26.3 | 3 | 260 |
| While using the product, did everything turn out to be what you had expected? Please also state what turned out to be expected/unexpected. | 25.8 | 7 | 285 |
| **Total** | 190.2 | 68 | 1821 |

Table 1: The values refer to the number of tokens across all participants' responses to the various prompts.

called expensive product/service. The distribution of emotions helps us to understand what kind of emotions are felt during the usage of an expensive product/service. On top of providing the list of eight emotions for participants to select, we also included a free text component for participants to write down the dominant emotion felt. This is for cases when participants did not endorse any of the eight emotions provided. Joy and disappointment appear to be the largest dominant positive and negative emotions felt, respectively.

Next, since each emotion is proposed to have a unique appraisal profile according to cognitive appraisal theory, we conducted a MANOVA on the 20 appraisal ratings to observe whether there are

| Emotion | % |
|---|---|
| Anger | 4.8 |
| Disappointment | 32.2 |
| Disgust | 2.0 |
| Gratitude | 11.3 |
| Joy | 19.2 |
| Pride | 4.6 |
| Regret | 14.4 |
| Surprise | 1.7 |
| Others | 9.7 |

Table 2: Distribution of emotion classes in the PEACE-Reviews dataset. These are the dominant emotions endorsed by the participants when recalling a specific experience.

differences between the eight emotions. We group all observations based on the dominant emotion selected by the participants. The omnibus MANOVA shows that there are significant differences amongst the eight emotion groups based on the 20 appraisal ratings, Wilk's $\Lambda$ is $F(140, 5358.68) = 8.86$, $p < .05$. We also conduct linear discriminant analyses that extend MANOVA to visualize the differences amongst emotion groups using appraisals. Figure 4 shows the boxplots of the first linear discriminant of appraisals for different emotions. We can observe that this discriminant is a function of the valence of emotions where emotions within the positive (gratitude, joy, pride) and negative (anger, disappointment, disgust, regret) emotion groups are similar. Surprise is considered to have both positive and negative valences.

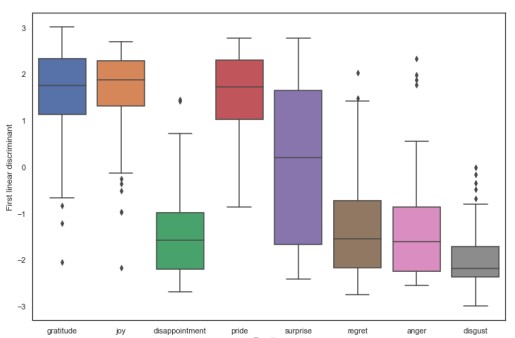

Figure 4: Boxplots of the first linear discriminant function of emotions.

# 7 Baseline Experiments

## 7.1 Tasks

First, we conducted an eight-way emotion classification task to predict the emotions of the combined text responses to the prompts. The second task also uses the combined text responses to the prompts

to predict each of the appraisal dimensions into 3 classes- high, medium, and low, typical of how psychologists study appraisals (Ellsworth and Smith, 1988; Scherer and Wallbott, 1994).

## 7.2 Baseline Models

We used the following baseline models to run the experiments-

- DistillRoBERTa (Hartmann, 2022), a transformer-based model, was fined tuned on a set of six emotion datasets such as GoEmotions (Demszky et al., 2020) and SemEval-2018 (Mohammad et al., 2018). This model contains six layers, 768 dimensions, and 12 attention heads.
- BERT (Devlin et al., 2018), a transformer model that applies bidirectional training to learn informative contextual features from sequences. We used the base model that contains 12 layers and attention heads, and 768 dimensions.
- XLNet (Yang et al., 2019) builds upon the BERT's limitations of failing to account for the dependencies between masked positions in sequences and is based on autoregressive language modeling that can learn bidirectional context. We used the base model that contains the same number of layers, attention heads, and dimensions as BERT.
- OpenAI's GPT3-ada (OpenAI, 2023) is a GPT-3-based model (Brown et al., 2020). We ran the model using the OpenAI API.

The dataset was split into training (80%), validation (10%), and test sets (10%), where the training and validation sets are used in finetuning to the respective tasks. The last layer of the models was modified accordingly to suit the number of classification outputs for the respective tasks. All models are fine-tuned on 10 epochs and implemented in Python.

## 7.3 Emotion prediction

For the emotion and appraisal prediction tasks, we use the classification accuracy and average weighted F1 scores as the evaluation metrics. Table 3 presents the results of the eight-way emotion classification. OpenAI's ada model appears to have the best results, and it even performs better than DistilRoBERTa which was pre-trained on multiple emotion corpus. Nevertheless, the performances of all of the models are comparable given that our

| Model | Accuracy | F1 |
|---|---|---|
| BERT | 67.9 | 0.62 |
| XLNet | 69.3 | 0.67 |
| DistillRoBERTa | 68.6 | 0.67 |
| OpenAI ada | 72.1 | 0.71 |

Table 3: Results of eight-way emotion classification. The values in the 'Accuracy' column are in percentages.

dataset has imbalanced emotion classes, with joy and disappointment dominating. Table 12 of Appendix B presents the accuracies and F1 scores for the individual emotions. As anticipated, since the emotions of joy and disappointment are the most dominant emotions in our dataset, the models excel in predicting these emotions but occasionally misclassify them into other positive and negative emotion classes. This suggests models primarily capture valence and pleasantness but struggle to differentiate specific emotions within each positive and negative emotion class.

Given the unique emotion classes in our dataset, direct comparisons with other datasets are challenging. However, a close comparison is the ISEAR dataset (Scherer and Wallbott, 1994), which has 3 emotions that overlapped with our set of 8 emotions, where BERT and XLNet achieved 70.1% and 73.0% accuracy in seven-way emotion classification, respectively (Adoma et al., 2020).

### 7.4 Appraisal prediction

Table 4 shows the results of the three-way appraisal classification for the appraisals of *goal conduciveness*, *goal relevance*, *pleasantness*, and *expectedness*. Since these are the appraisals that have text responses, we prioritize them and present them in the main text. The predictions of the other appraisals and the distribution of appraisal classes can be found in Table 14 and Table 13 of Appendix B. Across the three appraisals, the prediction accuracies of models on expectedness are the lowest. Unlike the appraisals of pleasantness and goal congruence, where positive/negative valence words could be easily identified and thus enhance the predictive performance, expectedness is a more complicated appraisal where participants described whether the experience was expected or unexpected. Even though DistillRoBERTa was pre-trained on emotion multiple emotion corpora, it did not have an edge in predicting the various appraisal dimensions except for *pleasantness*, which is expected since pre-training on multiple emotion corpora might learn words that are related to emotional valence.

| Appraisal/Model | Accuracy | F1 |
|---|---|---|
| **Pleasantness** | | |
| BERT | 73.4 | 0.74 |
| XLNet | 73.4 | 0.73 |
| DistillRoBERTa | 74.8 | 0.75 |
| OpenAI ada | 69.8 | 0.69 |
| **Goal conduciveness** | | |
| BERT | 70.5 | 0.71 |
| XLNet | 71.2 | 0.70 |
| DistillRoBERTa | 70.5 | 0.70 |
| OpenAI ada | 73.0 | 0.73 |
| **Expectedness** | | |
| BERT | 50.8 | 0.49 |
| XLNet | 57.8 | 0.55 |
| DistillRoBERTa | 55.5 | 0.53 |
| OpenAI ada | 56.1 | 53.7 |
| **Goal relevance** | | |
| BERT | 68.3 | 0.68 |
| XLNet | 73.5 | 0.73 |
| DistillRoBERTa | 69.1 | 0.68 |
| OpenAI ada | 66.1 | 0.65 |

Table 4: Results of three-way (high, medium, low) appraisal classification for *goal conduciveness*, *goal relevance*, *pleasantness*, and *expectedness*.

Overall, the models are comparably similar in the prediction of appraisals except for BERT's prediction for expectedness.

## 8 Conclusion

We curated a text dataset that described autobiographical expressions of participants' experiences when using an expensive product/service, annotated with a large number of self-reported psychological variables such as cognitive appraisals, emotions, traits, and future consumer behavior. We experimented with various writing prompts in the pilot study so as to determine which method elicits the best expression of emotional and cognitive appraisals of consumption experiences. Furthermore, we conducted experiments with state-of-the-art language models to provide baseline evaluations of appraisal and emotion classifications. Although we primarily examined appraisals and emotions in this paper, future research could utilize the other variables in this dataset such as decision-making traits, and post-purchase consumer behavior (e.g. intention to purchase and product again, and recommending the product to someone else) to model how appraisals and emotions interact with these variables. In summary, the PEACE-Reviews dataset enables researchers to study the correlates between expressed linguistic features and cognitive-emotional processes and affords a comprehensive understanding of emotional experiences in the domain of consumption and product/service reviews.

## Limitations

Although this curated dataset contains autobiographical text describing participants' appraisals and its associated ratings, we only included four appraisal questions (goal relevance, goal conduciveness, expectedness, and pleasantness) to elicit text responses about appraisals. Nevertheless, we selected these four appraisals because they are most relevant to the context of consumption (Desmet, 2008) and we feel that introducing more appraisal questions for participants' to complete could results in poor quality responses due to fatigue. Despite this, on top of the four appraisal questions, participants also provided text responses on questions such as 'what about the product/service that made you feel this way?' and 'why did you feel this way when you use the product/service?' alongside an appraisal questionnaire that contains 20 different appraisals. From the text responses elicited from these two questions and its associated appraisal ratings, linguistic features of appraisal dimensions could still be extracted.

## Ethics Statement

### 8.1 Composition

This dataset consists of participants' review text responses and the self-reported first-person ratings of emotion variables. These are the primary fields that are of interest to us. We also included an optional demographic questionnaire at the end of our survey. This demographic questionnaire was taken from Qualtrics's databases of demographic questionnaire templates that apply to the context of the US population. These questions are typically asked in research to determine the sample's composition. There was no personal data collected during the process of constructing this dataset.

### 8.2 Data Collection

The Departmental Ethics Review Committee of the Department of Communications and New Media at the National University of Singapore reviewed this study and determined it to be exempt. We respected the autonomy and protected the participants' privacy during every stage of the data collection process. Informed consent was obtained before the start of the study, and participants could withdraw at any point in time, including after providing their data, which we have discarded and not used in any subsequent analyses.

**Acknowledgments:** We thank Mike Cheung, Harold Soh, and Desmond Ong for early feedback on the research design. This work was supported by a grant from the NUS Centre for Trusted Internet and Community.

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

# A Supplementary Information for Pilot Study

Table 5 provides information about the prompts and questions used across different conditions. The "Recollection phase" is when participants are recalling an experience of using an expensive product/service. During the "Writing phase", participants then write down their experiences according to the questions asked in the different conditions.

| Condition | Recollection phase | Writing phase |
|---|---|---|
| **1 (Emotion + Review)** | Think about the time when you felt happy/angry when you were using the expensive product/service that you have thought about. Picture this situation in your mind. Try your best to remember this situation as vividly as you can. | After you have thought about your experience, **describe this moment in the form of a review** of the recalled product/service as if you are going to publish this review online. Please write in complete sentences and as detailed as possible. |
| **2 (Emotion + Questionnaire)** | –same as above– | After you have thought about your experience, **describe this moment in the following questions,** 1. Was the product/service important to you? Why? 2. What about the product that made you feel happy/angry? 3. Why were you happy/angry when you used or experienced the product? 4. Was the product/service consistent with what you wanted? Why? 5. Was it a pleasant experience when you were using the product/service? Why? 6. While using the product, did everything turn out to be what you had expected? Why? |
| **3 (Review)** | Please think about the time when you were using the expensive product/service that you have thought about. Picture this situation in your mind. Try your best to remember this situation as vividly as you can. | After you have thought about your experience, **describe this moment in the form of a review** of the recalled product/service as if you are going to publish this review online. Please write in complete sentences and as detailed as possible. |
| **4 (Questionnaire)** | –same as above– | After you have thought about your experience, **describe this moment in the following questions,** 1. Was the product/service important to you? Why? 2. How did you feel when you used the product/service? Please also state what about the product/service that made you feel this way? 3. Was the product/service consistent with what you wanted? Why? 5. Was it a pleasant experience when you were using the product/service? Why? 6. While using the product, did everything turn out to be what you had expected? Why? |

Table 5: Prompts and questions for the different condition.

Table 6 presents the appraisal questionnaire used in both the pilot and main studies (Yeo and Ong, 2023). The third column contains values for the percentage of NA responses for each appraisal dimensions in the main study. As not all appraisal dimensions are relevant in the context of consumption, the percentage of NA responses allows us to discern which appraisals are important in consumption experiences. Table 7 presents the emotion questionnaire used in both the pilot and the main studies. Both the appraisal and emotion question-

naires and rated on a 7-point Likert scale.

| Appraisal | Measure | % of NA |
|---|---|---|
| Accountability-circumstances | To what extent did you think that circumstances beyond anyone's control were responsible for what was happening in the situation? | 16.43 |
| Accountability-other | To what extent did you think that someone else other than you was responsible for what was happening in the situation? | 11.79 |
| Accountability-self | To what extent did you think that you were responsible for what was happening in the situation? | 4.93 |
| Attentional activity | To what extent did you think that you needed to attend to the situation further? | 14.64 |
| Certainty | To what extent did you understand what was happening in the situation? | 5.14 |
| Control-circumstances | To what extent did you think that circumstances beyond anyone's control were controlling what was happening in the situation? | 14.14 |
| Control-other | To what extent did you think that other people were controlling what was happening in the situation? | 14.21 |
| Control-self | To what extent did you think you had control over the situation? | 2.07 |
| Coping potential | To what extent were you able to cope with any negative consequences of the situation? | 16.36 |
| Difficulty | To what extent did you think that the situation was difficult? | 6.79 |
| Effort | To what extent did you think that you needed to exert effort to deal with the situation? | 11.57 |
| Expectedness | To what extent did you expect the situation to occur? | 9.00 |
| External normative significance | To what extent did you think that the situation was consistent with external and social norms? | 15.50 |
| Fairness | To what extent did you think the situation was fair? | 5.71 |
| Future expectancy | To what extent did you think that the situation would get worse/better? | 13.21 |
| Goal conduciveness | To what extent was the situation consistent with what you wanted? | 0.86 |
| Goal relevance | To what extent did you think that the situation was relevant to what you wanted? | 3.14 |
| Novelty | To what extent did you think that the situation was familiar? | 12.07 |
| Perceived obstacle | To what extent did you think that there were problems that had to be solved before you could get what you wanted? | 11.57 |
| Pleasantness | To what extent did you think that the situation was pleasant? | 0.71 |

Table 6: The cognitive appraisal dimensions measured in this dataset. The questionnaire is sourced from prior work (Scherer et al., 1984; Smith and Ellsworth, 1985; Tong and Jia, 2017). The third column contains values for the percentage of NA responses for each appraisal obtained in the main study.

We present some sample responses of the questionnaire and review formats in Table 8 and Table 9, respectively. In general, the questionnaire format

| Emotion | Measure |
|---|---|
| Anger | To what extent did you feel angry? |
| Disappointment | To what extent did you feel disappointed? |
| Disgust | To what extent did you feel disgusted? |
| Gratitude | To what extent did you feel grateful? |
| Joy | To what extent did you feel happy? |
| Pride | To what extent did you feel proud? |
| Regret | To what extent did you feel regretful? |
| Surprised | To what extent did you feel surprised? |

Table 7: The emotion dimensions measured in this dataset on a 7-point Likert scale. The questionnaire is sourced from prior work (Scherer et al., 1984; Smith and Ellsworth, 1985; Tong and Jia, 2017)

resulted in higher quality responses in terms of the expression of cognitive appraisals and emotions in the text.

| Question | Condition 2 | Condition 4 |
|---|---|---|
| Was the product/service important to you? Please also state the reason(s) of why was the product/service important/unimportant to you. | *It was important as a part of self care; I'd never spend such an amount on just my own enjoyment before, and it was liberating. There were no corners cut at all regarding the experience, which was a three day stay at a very exclusive resort.* (Joy) | *Yes because this product was important to me since it was an expensive meal with my friends. I have good memories of this experience, and I got to try new things that I have not tried before.* |
| How did you feel when you used the product/service? Please also state what about the product/service that made you feel this way? | *It was best described as a constant state of bewilderment at being pampered as I was. It was elaborately exclusive and luxurious, with amazing views and amenities. If not for the lousy weather, it would have been perfect, but even then I appreciated the scenery in falling rain.* (Joy) | *I felt happy and I felt that I was eating one of the best meals of my life. The product was excellent sushi chosen by the sushi chef, and it had just the right amount of wasabi and soy sauce on each piece.* |
| Was the product/service consistent with what you wanted? Please also state the reason(s) of why was the product/service consistent with what you wanted. | *It was very consistent with what I wanted; I wanted the absolute pinnacle of a resort experience in terms of exclusivity and comfort and care and detail, and the Intercontinental provided that from landing to departure. I never lacked for any service and everything was consistently excellent.* (Joy) | *Yes because I expected great sushi for the high price, and it was the best sushi that I had in my life. Therefore, the product was consistent with what I expected/wanted.* |
| Was it a pleasant experience when you were using the product/service? Please also state the reason(s) of why was it a pleasant experience when you were using the product/service. | *Yes because it was very pleasant; every detail of the experience was expertly curated and I could note no meaning fault in any of it. The food was exquisite, the views were incredible, the beds comfortable and the amenities luxurious.* (Joy) | *Yes because I had a good time with my friends, and it was a pleasant dining experience as well as just a good time with my friends.* |

Table 8: Examples of review text responses for the question format conditions (condition 2-presence of emotion prompts, condition 4- absence of emotion prompts). The general emotion that the participants felt when using the product is in parentheses for condition 1.

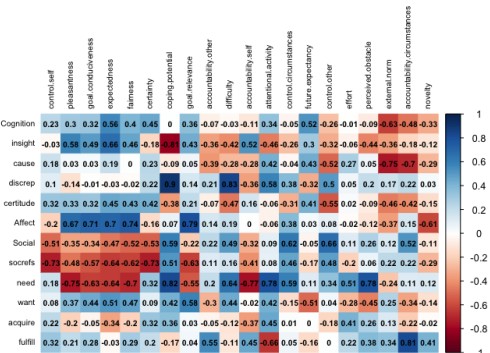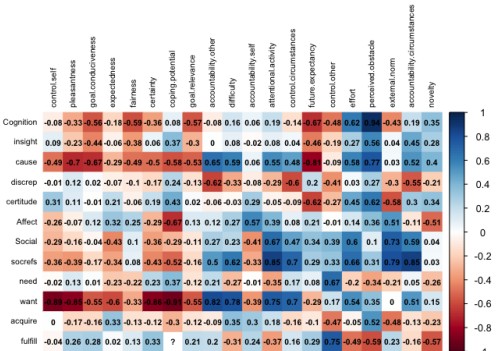

Figure 5: Correlation between selected LIWC-22 dimensions and the appraisal dimensions in the current study. The left and right panel represent the correlations for the responses of the questionnaire and review formats, respectively. The vertical axis labels are the LIWC-22 dimensions, while the horizontal axis labels are the appraisal dimensions in our study. "discrep" represents "discrepancy," and "socref" represents "social referents." The full description of the categories, their definitions, and contents are available in the LIWC 2022 manual.[3]

| Emotion | Condition 1 | Condition 3 |
|---------|-------------|-------------|
| Joy | I love my Toyota Camry! The gas mileage is top notch. I can sometimes travel to and from work for two weeks before I have to fill up. When I do fill up it doesn't break the bank. | The service provided was exceptional. We were treated well and the premises were pristine. Would do it again! |
| Anger | I bought this product hoping to better my self-tape procedures, since I am an actress. However, the product was defective from the very beginning and difficult to wok with. The entire product broke a few weeks after. | We were lucky enough to be seated right away because of a cancelation. The service was excellent and the food was amazing,by far the freshest,best tasting food we have had in this area in awhile. The food had an up-scale feel without the upscale price tag. |

Table 9: Examples of review text responses for the review format conditions (condition 1-presence of emotion prompts, condition 3- absence of emotion prompts). The self-reported general emotion that the participants felt when using the product is in column 1.

Figure 5 presents the correlations between selected LIWC22 dimensions and the 20 appraisal dimensions for the questionnaire and review examined in the current study. Although there are no validated appraisal lexicon in the literature, certain LIWC-22 dimensions might be useful in indicating some of the cognitive appraisals. For instance, the dimensions of *certitude* and *insight* are indicative of the appraisal dimension *certainty* and the correlation of these LIWC22 dimensions and *certainty* are greater for the questionnaire compared to the review format. The dimensions of *causation*, *social*, and *socrefs* might be indicative of cognitive appraisals of *accountability* and *control* which the questionnaire format has in general a high correlation between these dimensions compared to the review format. Also, the questionnaire format generally has high correlations between dimensions of *need*, and *want* with the cognitive appraisal of *goal*

*relevance*. The full description of the categories, their definitions, and contents are available in the LIWC 2022 manual.[4]

# B Supplementary Information for Main Study

We collected an optional demographics questionnaire at the end of the main study. Table 10 presents the distribution of participants across different demographic variables. Do note that we only sampled participants from the United States to reduce the heterogeneity of the responses collected.

Figure 6 shows the correlation between a subset of all the measured variables in our study. From these correlations, we observe that different specific emotions have different relationships to different appraisals and positive emotions positively correlate with intention to recommend to others and purchase again.

We present the means of the appraisals with respect to the eight emotions in Table 11. Most of the relationships are generally consistent with previous theories (Yeo and Ong, 2023). For example, anger is associated with high scores on *accountability-other*, *perceived obstacle*, *pleasantness*, and low scores on *external normative significance*, *fairness*, *goal conduciveness*. Pride is associated with high scores on *accountability-self*, *certainty*, *control-self*, *goal conduciveness*, *pleasantness*. However there are some appraisal-emotion relationships that are not consistent. For example, surprise is hypothesized to be associated with high *attentional activity* but in our study we found that it has a low

---
[4]Available at https://tinyurl.com/liwc2022-manual

| N | 700 |
|---|---|
| **Gender** | female: 42.3% |
| | male: 54.9% |
| | Non-binary/third gender: 2.3% |
| | no response: 0.6% |
| **Age group** | 18-24 years: 9.1% |
| | 25-34 years: 29.6% |
| | 35-44 years: 22.3% |
| | 45-54 years: 15.1% |
| | 55-64 years: 14.9% |
| | 64 years: 9.1% |
| **Race** | White or Caucasian: 63.3% |
| | Black or African American: 22.6% |
| | American Indian/Native American: 2.6% |
| | Asian: 13.3% |
| | Others: 2.0% |
| | no response: 1.0% |
| **Hispanic/Latino origin** | Yes: 12.0 % |
| | No: 86.6% |
| | no response: 1.4% |
| **Highest education level** | High school diploma or less: 12.7% |
| | Technical degree: 10.3% |
| | Some college but no degree: 16.9% |
| | Bachelor Degree: 37.9% |
| | Graduate degree: 22.0% |
| | no response: 0.3% |
| **Annual household income** | less than $25,000: 13.7% |
| | $25,000-$49,999: 22.4% |
| | $50,000-$79,999: 18.0% |
| | $75,000-$99,999 - 14.1% |
| | $100,000-$149,999 - 18.0% |
| | $150,000 or more - %11.0 |
| | no response: 2.7% |

Table 10: Demographic statistics of participants in the main study.

rating. Gratitude is hypothesized to be associated with high *accountability-other* but we found here that it has a low rating.

Figure 7 shows an example of how specific emotions (e.g. joy and disappointment) have different appraisal profiles. The emotion of joy is associated with high pleasantness, goal conduciveness, and moderate expectedness, while disappointment is a negative emotion that is low in pleasantness, goal conduciveness, and expectedness.

Table 12 presents the performance accuracies and F1-scores for individual emotions concerning the different models. Across all the models, the accuracies for the emotions of *disappointment*, *gratitude*, and *joy* were the highest. This is expected as these emotions are the most dominant emotions endorsed by the participants in our dataset. The models perform poorly in predicting the least endorsed emotions such as *disgust* and *surprise* as there are not many instances of such emotions.

Table 13 shows the distribution of the cognitive appraisals in terms of low, medium, and high used in the appraisal prediction task. In general, most of the distributions across the three classes are balanced except for *accountability-circumstances*, *control-circumstances*, *difficulty*, *expectedness*, and *novelty*, which have very little instances in the "high" class (less than 10%).

Table 14 shows the three-way prediction results of the other 16 appraisal dimensions. We can observe that certain appraisals such as *accountability-self*, *control-circumstances*, *control-other*, *control-self*, *difficulty*, *external normative significance*, and *fairness* have prediction results that have at least one model predicting above an accuracy of 60%. This suggests that these mentioned appraisals exhibit certain systematic linguistic features that models readily pick up and use it for predictions, compared to other appraisals. The appraisal with the lowest accuracy is *accountability-other* which is surprising because we expected that the models are able to pick up linguistic features of attribution and responsibility of the situation, nouns and pronouns (e.g. *waiter*, *service staff*, *they*) which might be indicative of this appraisal.

| Appraisal | Anger | Disappointment | Disgust | Gratitude | Joy | Pride | Regret | Surprise |
|---|---|---|---|---|---|---|---|---|
| Accountability-circumstances | 3.28 | 3.28 | 2.92 | 2.66 | 2.74 | 3.02 | 3.07 | 2.80 |
| Accountability-other | 5.43 | 4.53 | 4.89 | 3.95 | 3.64 | 3.07 | 3.79 | 4.48 |
| Accountability-self | 2.83 | 2.72 | 3.36 | 5.24 | 5.27 | 5.73 | 4.31 | 3.83 |
| Attentional actvitiy | 5.14 | 4.46 | 4.65 | 2.73 | 2.72 | 3.23 | 4.12 | 3.60 |
| Certainty | 4.52 | 4.48 | 4.43 | 6.51 | 6.44 | 6.08 | 4.96 | 5.87 |
| Control-circumstances | 2.89 | 3.19 | 3.46 | 2.74 | 2.74 | 3.11 | 3.08 | 3.24 |
| Control-other | 4.95 | 3.89 | 4.72 | 3.02 | 2.93 | 2.85 | 3.45 | 4.10 |
| Control-self | 3.15 | 3.03 | 3.64 | 5.82 | 5.65 | 6.09 | 4.19 | 4.38 |
| Coping potential | 4.18 | 4.39 | 4.22 | 5.94 | 5.70 | 5.38 | 4.50 | 4.67 |
| Difficult | 5.12 | 4.45 | 5.04 | 1.81 | 1.92 | 2.39 | 4.27 | 2.55 |
| Effort | 5.45 | 4.54 | 4.88 | 2.92 | 2.81 | 3.45 | 4.53 | 3.32 |
| Expectedness | 3.09 | 3.06 | 3.26 | 3.87 | 3.61 | 4.21 | 3.23 | 3.18 |
| External normative significance | 3.43 | 3.54 | 3.36 | 5.42 | 5.71 | 5.50 | 4.06 | 4.43 |
| Fairness | 2.49 | 2.66 | 3.36 | 6.27 | 6.22 | 6.00 | 3.56 | 5.13 |
| Future expectancy | 3.81 | 3.58 | 3.62 | 5.39 | 5.33 | 5.50 | 3.57 | 4.95 |
| Goal conduciveness | 2.43 | 2.27 | 3.07 | 6.45 | 6.46 | 6.34 | 3.37 | 4.88 |
| Goal relevance | 2.86 | 2.92 | 3.64 | 6.51 | 6.46 | 6.34 | 3.89 | 5.50 |
| Novelty | 3.06 | 2.89 | 3.22 | 3.97 | 4.04 | 4.53 | 3.22 | 4.05 |
| Perceived obstacle | 5.35 | 4.71 | 4.41 | 2.77 | 2.76 | 3.25 | 4.43 | 3.18 |
| Pleasantness | 2.46 | 2.54 | 3.18 | 6.46 | 6.46 | 6.34 | 3.49 | 5.13 |

Table 11: Means of the appraisal dimensions with respect to different emotions. These appraisals are rated on a 7-point Likert scale.

| Emotion | BERT | | XLNet | | DistillRoBERTa | |
|---|---|---|---|---|---|---|
| | Acc | F1 | Acc | F1 | Acc | F1 |
| Anger | 20.2 | 0.33 | 33.3 | 0.33 | 33.3 | 0.38 |
| Disappointment | 92.0 | 0.78 | 90.0 | 0.83 | 84.0 | 0.79 |
| Disgust | 0.00 | 0.00 | 0.00 | 0.00 | 14.3 | 0.25 |
| Gratitude | 80.0 | 0.67 | 80.0 | 0.69 | 80.0 | 0.71 |
| Joy | 91.3 | 0.76 | 87.0 | 0.74 | 78.3 | 0.75 |
| Pride | 25.0 | 0.40 | 50.0 | 0.67 | 62.5 | 0.56 |
| Regret | 54.5 | 0.67 | 50.0 | 0.63 | 54.5 | 0.59 |
| Surprise | 0.00 | 0.00 | 33.3 | 0.40 | 50.0 | 0.67 |

Table 12: Accuracy and F1 scores for each emotion class of each model.

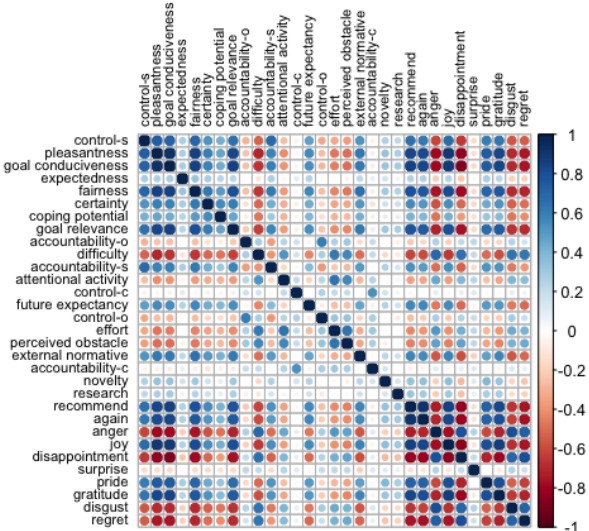

Figure 6: Correlation matrix of a subset of variables in the dataset. "control-s" refers to "control-self", "accountability-o" refers to "accountability-other", "accountability-s" refers to "accountability-self", "control-c" refers to "control-circumstances", "external normative" refers to "external normative significance", "accountability-c" refers to "accountability-circumstances", "research" refers to whether the participant put in effort in researching for the product/service before purchasing, "again" refers to whether the participant will purchase the product/service again.

| Appraisal | Low (%) | Medium (%) | High (%) |
|---|---|---|---|
| Accountability-circumstances | 59.4 | 34.4 | 6.2 |
| Accountability-other | 39.8 | 40.4 | 19.8 |
| Accountability-self | 40.4 | 4.93 | |
| Attentional activity | 43.4 | 42.6 | 14.0 |
| Certainty | 16.3 | 42.0 | 41.6 |
| Control-circumstances | 59.8 | 34.4 | 5.8 |
| Control-other | 50.7 | 38.2 | 11.1 |
| Control-self | 34.4 | 45.1 | 20.5 |
| Coping potential | 22.8 | 53.0 | 24.2 |
| Difficulty | 52.0 | 39.4 | 8.7 |
| Effort | 40.8 | 45.9 | 13.3 |
| Expectedness | 50.6 | 42.0 | 7.4 |
| External normative significance | 26.3 | 53.8 | 19.9 |
| Fairness | 37.0 | 35.6 | 27.4 |
| Future expectancy | 26.3 | 57.0 | 16.6 |
| Goal conduciveness | 40.4 | 29.9 | 29.7 |
| Goal relevance | 33.3 | 32.6 | 34.3 |
| Novelty | 48.4 | 42.5 | 9.1 |
| Perceived obstacle | 42.7 | 41.6 | 15.7 |
| Pleasantness | 37.2 | 32.8 | 30.0 |

Table 13: Distribution of cognitive appraisals in terms of low, medium, and high used in the appraisal prediction task.

Figure 7: Comparing different appraisal ratings across the emotion of 'joy' and 'disappointment.'

| Appraisal | BERT | | XLNet | | DistillRoBERTa | | OpenAI ada | |
|---|---|---|---|---|---|---|---|---|
| | Acc | F1 | Acc | F1 | Acc | F1 | Acc | F1 |
| Accountability-circumstances | 50.4 | 0.45 | 47.0 | 44.5 | 53.0 | 0.48 | 57.7 | 0.55 |
| Accountability-other | 45.2 | 0.43 | 46.8 | 0.46 | 35.5 | 0.35 | 44.9 | 0.44 |
| Accountability-self | 53.7 | 0.54 | 50.0 | 0.50 | 56.0 | 0.56 | 61.3 | 0.61 |
| Attentional activity | 48.3 | 0.44 | 45.0 | 0.42 | 51.7 | 0.51 | 56.5 | 0.54 |
| Certainty | 55.6 | 0.55 | 56.4 | 0.56 | 52.6 | 0.52 | 50.4 | 0.50 |
| Control-circumstances | 64.5 | 0.58 | 65.3 | 0.52 | 63.6 | 0.61 | 55.0 | 0.54 |
| Control-other | 62.0 | 0.60 | 54.5 | 0.55 | 52.9 | 0.54 | 55.0 | 0.53 |
| Control-self | 61.6 | 0.60 | 59.4 | 0.58 | 60.1 | 0.59 | 65.0 | 0.65 |
| Coping potential | 51.7 | 0.45 | 45.8 | 0.38 | 44.1 | 0.43 | 57.7 | 0.55 |
| Difficulty | 67.1 | 0.67 | 64.9 | 0.63 | 70.2 | 0.69 | 64.8 | 0.62 |
| Effort | 54.8 | 0.51 | 56.4 | 0.53 | 58.9 | 0.57 | 55.6 | 0.53 |
| External normative significance | 64.7 | 0.63 | 59.7 | 0.59 | 63.0 | 0.62 | 53.6 | 0.54 |
| Fairness | 66.7 | 0.66 | 68.2 | 0.66 | 64.4 | 0.64 | 68.2 | 0.68 |
| Future expectancy | 54.9 | 0.54 | 58.2 | 0.53 | 59.0 | 0.55 | 58.4 | 0.57 |
| Novelty | 57.3 | 0.54 | 56.5 | 0.53 | 56.5 | 0.53 | 61.8 | 0.60 |
| Perceived obstacle | 58.9 | 0.54 | 54.0 | 0.52 | 48.4 | 0.57 | 52.8 | 0.53 |

Table 14: Results of three-way (high, medium, low) appraisal classification for the other 16 appraisal dimensions.