# OpenReview forum: "The PEACE-Reviews dataset: Modeling Cognitive Appraisals in Emotion Text Analysis"
_EMNLP/2023/Conference — EMNLP 2023 Findings_

### Official Review · Reviewer_Miqh · 2023-08-01

**Typos Grammar Style And Presentation Improvements:** 1. Line 397-398
**Soundness:** 3

**Excitement:**

3: Ambivalent: It has merits (e.g., it reports state-of-the-art results, the idea is nice), but there are key weaknesses (e.g., it describes incremental work), and it can significantly benefit from another round of revision. However, I won't object to accepting it if my co-reviewers champion it.

**Paper Topic And Main Contributions:**

This study introduced a novel dataset, PEACE-Reviews, an autobiographical writing review collection, designed to tackle the limitations observed in currently available public datasets:

1. Many existing datasets lack annotations for both emotions and cognitive appraisals, hindering the study of the psychological connection between them.
2. Existing datasets often fall short in capturing a rich variety of emotional language and labels.
3. Present datasets predominantly reflect a third-person's perception of the written text, rather than the original author's (first-person) appraisals.

Additionally, baseline experiments were conducted on this dataset to identify emotions and cognitive appraisals utilizing linguistic features.

**Questions For The Authors:**

A. The term “expensive” can be somewhat ambiguous. One might assume that people tend to be more critical of pricier products or services and more easily pleased with economical ones.

B. Many individuals, if not the majority, tend to be more introspective and less emotionally charged when reflecting on past experiences, having had time to process. Have you considered this factor? Furthermore, the recency of an experience can influence recall and emotion. Did you set guidelines regarding how recent these experiences should be?

C. Is there a quantitative analysis supporting the conclusions/observations made in section 5.2?

D. While it seems intuitive that a questionnaire format might yield more detailed results due to its structured nature, what is the core discovery here? Is it that structured guidance elicits richer responses? If the concern is that review format responses tend to emphasize product specifics (as noted in lines 361-362), you could potentially direct participants to concentrate on their emotional reactions and thought processes rather than product details.

E. Did you measure any time discrepancies between completing a questionnaire-based survey versus a review-based one?

F. How do you differentiate between functional words (e.g., the word “get” in sentences like "This gets better") and genuine psychological state lexicons (e.g., "I got this product")?

G. How do you ensure authenticity in the responses? Is there a mechanism to detect fabricated stories?

H. Are your dataset's statistics consistent with the proposed emotion-appraisal profiles? Specifically, do you identify which appraisal dimensions rank high or low for particular emotions, and how these compare to existing theories?

I. Is there a theoretical or quantitative basis for selecting the four appraisal questions to elicit text responses about appraisals? At a glance, I perceive that factors like customer service quality, associated with attributes such as “other-accountability” and “difficulty”, might also influence emotions.

J. In appraisal prediction experiments, why were only the appraisal dimensions of “pleasantness”, “goal conduciveness”, and “expectedness” considered? In particular, why was “goal relevance” excluded, especially since it was one of the prompt queries?

**Reasons To Accept:**

This study explores the nuanced connection between self-cognitive appraisal and its consequent effects on emotions and behaviors. By probing into the feasibility of extracting self-cognitive appraisal from autobiographical writing reviews, the author(s) contribute a unique and insightful perspective to the field. The proposed novel dataset could serve as a valuable asset for further research in this domain.

**Reasons To Reject:**

The rationale for selecting the specific topic (expensive product/service purchase experience) requires further justification (refer to question A below). Additionally, the dataset seems to lack a quantitative analysis (see questions C and H). The baseline experiments appear preliminary. It would be beneficial to see experiments exploring if cognitive appraisal inference/modeling enhances emotion detection, and potentially, future customer behaviors, especially since data regarding intentions of future purchases and recommendations has been gathered.

**Reproducibility:**

4: Could mostly reproduce the results, but there may be some variation because of sample variance or minor variations in their interpretation of the protocol or method.

**Reviewer Confidence:**

4: Quite sure. I tried to check the important points carefully. It's unlikely, though conceivable, that I missed something that should affect my ratings.

---

> ### Author Rebuttal · Authors · 2023-08-28
>
> We are extremely grateful for your valuable feedback. We'll be sure to increase the size of our tables and figures in the camera-ready version of the paper. As we mention in the "Summary of Rebuttal" comment, the research design and the dataset collection are the main contributions of the paper. The choice to report only simple predictive deep-learning baselines on the final dataset was deliberate so that we could focus on our unique contribution. We plan to explore directions in predictive language and behavioral modeling in subsequent projects, while also opening this dataset up for other researchers to use.
>
> Answer A: Thank you for the great observation! Anticipating these differences, we also asked the respondents how much they spent on the particular product so that we could control for these factors in subsequent analyses. This information is included in the dataset. Nevertheless, despite these differences, we were motivated by our goal of curating a dataset that is rich in emotional expressions. According to Richins (2008), this class of products has the highest likelihood of experiencing an emotion and also results in higher emotional intensity while using it as its usage represents a situation that the participants think deeply about before purchasing and is personally important to them. Compared to someone using a mundane or economical product such as a pen or a toothbrush, expensive products (regardless of whether the participant is critical of it or not, i.e. think it is not worth it or regret buying it) are able to increase the likelihood of experiencing an emotion and the intensity of felt emotions.
>
> Answer B: The design of the data collection closely followed the methodological frameworks of previous studies that investigate appraisals and emotions in Psychology (Scherer, 1997; Smith & Ellsworth, 1985; Tong, 2015). We tried our best to facilitate recall by including prompts such as ‘How did you feel when you used the product/service?’ and ‘What about the product that caused you to feel the emotion stated?’. Moreover, prior to the main study, we also conducted screening studies to ensure that participants were able to recall an emotional episode of using an expensive product/service. Similar to Answer A, we feel that the context of using an expensive product/service helps to facilitate recall and ensure that rich emotional experiences are collected.
>
> Answer C: The conclusion and observations were based on a few quantitative analyses. Firstly, in terms of how detailed the responses were, we used a word count approach to analyze how detailed the text responses were. For example, the average word count for a text of the review format is 61.8 words whereas the average word count for a text of the questionnaire format is 150 words. Since we are interested in understanding appraisals from such text, a detailed account would be preferable as it has the potential to extract various appraisal dimensions from the text. Secondly, in terms of whether the appraisal ratings are consistent with the expressed responses, we coded for several appraisal dimensions using LIWC and correlated them with the corresponding appraisal ratings. We will include these findings in the revised paper.
>
> Answer D: One of the main concerns when designing the data collection procedure is that the review format responses might emphasize product-specific details instead of emotional reactions even when we ask participants to write about the emotional aspect of it. Another primary concern is that the review format responses might not contain text pertaining to appraisal dimensions. Therefore, we carried out the pilot study to determine whether a questionnaire format might yield more detailed emotional responses or whether a review format is as emotionally detailed as the questionnaire format. From the results, we observed that indeed the structured nature of the questionnaire format elicits richer emotional responses that contain appraisal dimensions that could be used to analyze such text.
>
> Answer E: Yes since we have data on how long each participant completed the tasks, we found that the review-based task took an average of 11.15 minutes while the questionnaire-based task took an average of 13.70 minutes to complete.
>
> Answer F: We did not differentiate between functional and genuine psychological state lexicons. Since there are no appraisal lexicons in the literature, we used some of the psychological state lexicons from LIWC as a proxy for certain appraisal dimensions. For example, the appraisal of ‘goal conduciveness’ could be measured by a combination of various psychological states such as ‘get’, ‘obtain’, and ‘fulfill’. Therefore, the aggregation of various psychological states might mitigate the presence of functional words in the dataset.
>
> Answer G: We could also ensure that the responses are written by real humans instead of bots due to the validation and vetting of participants in the Prolific platform. The Prolific platform is also recently working on various ways to identify participants who used artificial intelligence such as ChatGPT to respond to tasks. However, we did not implement any mechanism to detect fabricated stories by human participants.
>
> Answer H: Due to space constraints and the scope of the paper, we only presented preliminary results on how the ratings of appraisals are related to the eight included emotions (see line 521-541). From the MANOVA results, we found that there are significant differences amongst the eight emotion groups based on the 20 appraisal ratings. However, post-hoc analyses to observe which appraisals rank high or low for particular emotions are not presented. We are planning to analyze the appraisal-emotion profiles in another separate paper. Nevertheless, we could present these results in the Appendix of the revised paper.
>
> Answer I: We agree that other appraisal dimensions might be relevant in the context of consumption but due to the resources constraints and feasibility of data collection, we only collected text responses for four specific appraisal questions - 1. goal relevance, 2. goal conduciveness, 3. expectedness, and 4. pleasantness, instead of eliciting responses for all 20 appraisal dimensions. The choice of these appraisal dimensions to elicit text responses is supported by previous studies that show certain appraisals that are particularly relevant to the context of consumption (Demir et al., 2009; Desmet, 2008; Desmet & Hekkert, 2007). Nevertheless, we also added two questions that might elicit other appraisals (see Table 8- a) How did you feel when you used the product/service?, and b) What about the product/service that made you feel this way?).
>
> Answer J: For the selection of appraisal dimensions used for the classification task, we only included the appraisal dimensions that have text responses that are elicited from the appraisal questions (i.e. goal relevance, goal conduciveness, expectedness, and pleasantness). Due to space constraints, we only presented goal conduciveness, expectedness, and pleasantness for the appraisal classification tasks. The primary purpose of the appraisal classification task is to demonstrate that the text and ratings are useful in measuring and understanding appraisals in consumption. In the revised version, we will include the classification of goal relevance and other appraisal dimensions for completeness.
>
> References
>
> Desmet, P., & Hekkert, P. (2007). Framework of product experience. International journal of design, 1(1), 57-66.
>
> Desmet, P. M. (2008). Product emotion. In Product experience (pp. 379-397). Elsevier.
>
> Demir, E., Desmet, P. M., & Hekkert, P. (2009). Appraisal patterns of emotions in human-product interaction. International journal of design, 3(2). Richins, M. L. (2008). Consumption emotions. In Product experience (pp. 399-422). Elsevier.
>
> Scherer, K. (1997). Profiles of emotion-antecedent appraisal: Testing theoretical predictions across cultures. Cognition & Emotion, 11(2), 113-150.
>
> Smith, C. A., & Ellsworth, P. C. (1985). Patterns of cognitive appraisal in emotion. Journal of personality and social psychology, 48(4), 813.
>
> Tong, E. M. (2015). Differentiation of 13 positive emotions by appraisals. Cognition and Emotion, 29(3), 484-503.

---

### Official Review · Reviewer_rMKV · 2023-08-03

**Soundness:** 3

**Excitement:**

4: Strong: This paper deepens the understanding of some phenomenon or lowers the barriers to an existing research direction.

**Missing References:**

None

**Paper Topic And Main Contributions:**

This paper presents the Perception and Experience of Appraisals and Consumption Emotions in Reviews (PEACE) dataset, a collection of reviews annotated with first-person emotional variables, intended to be used to measure emotional states and cognitive appraisal dimensions in product and service reviews. This is the first dataset of such kind since existing datasets are only annotated according to third-person appraisal (and therefore model perception rather than experience).
To construct the dataset, the authors first ran a pilot study to identify the most effective writing prompt to use in the main study. For the main data collection, 700 participants were then recruited via Prolific and asked to rate their emotions (among an option pool of eight emotions) and appraisal (from a set of appraisal dimensions) experienced when using an expensive product or service that they recently purchased. The resulting dataset contains a total of 1400 examples (each participant provided two responses, one positive and one negative). The results show that joy and disappointment are the most dominant emotions selected by participants.
The paper then presents baseline experiments for predicting emotions and appraisal dimensions from text. To do so, they cast the problem as an 8-class classification task. Additionally, the authors aim to predict each of the appraisal dimensions as a 3-class classification task (for the appraisal dimensions goal conduciveness, pleasantness, and expectedness). Baseline models include DistillRoBERTa, BERT, XLNet, and GPT-3. Performance metrics are reported in terms of accuracy and F1, showing that GPT-3 provides the best results in terms of F1 for emotion classification. For appraisal classification, expectedness is hardest to predict, and performances are higher for the other two categories. The different models perform similarly across comparisons.

**Questions For The Authors:**

Did you run experiments with linear models for the classification of emotions and appraisal dimensions? This could be interesting to analyze feature importances of the model, which would provide potential insights into which words drive classification performance.

**Reasons To Accept:**

* The paper presents a novel dataset, with product/service reviews annotated according to first-person emotional and appraisal states, which represents the first of its kind in this context.
* The paper provides an extensive review of the literature to give context to the theory of cognitive appraisal.
* The dataset creation process is well-documented. The authors furthermore conducted pilot studies to identify to optimize their collection process, which speaks to the quality of the dataset.


**Reasons To Reject:**

* The selection of appraisal dimensions used for classification seems unclear. In lines 501-503, it is mentioned that goal conduciveness and pleasantness have the lowest number of NA responses. In Section 7.4, results are shown for a 3-class classification task for goal conduciveness, pleasantness, and expectedness. In the Limitations Section, it is mentioned that only four appraisal questions are used to elicit responses. However, Table 6 shows many more dimensions and the caption states that all these dimensions were measured in the dataset. Apologies if I am confusing this, but can you clarify and elaborate on how you selected those 3 (4?) categories and why you included them for the classification task?
* Related to the above, the distribution of class labels for the appraisal task (low-medium-high) does not seem to be shown in the paper. I would recommend adding this since it has an impact on how to interpret performance scores in Table 4.
* The analysis of linguistic features regarding the dataset could be more extensive. While the collection process is described in great detail, the data analysis is rather thin and could be addressed further (e.g., by providing results on word and n-gram occurrences across emotions and appraisal dimensions, as well as topic modeling). This would serve to provide the reader with a better understanding of how individual reviews are composed, and what words are indicative of individual emotional states and appraisal dimensions.


**Reproducibility:**

4: Could mostly reproduce the results, but there may be some variation because of sample variance or minor variations in their interpretation of the protocol or method.

**Reviewer Confidence:**

3: Pretty sure, but there's a chance I missed something. Although I have a good feel for this area in general, I did not carefully check the paper's details, e.g., the math, experimental design, or novelty.

**Typos Grammar Style And Presentation Improvements:**

* Line 218: “third-person” appearing twice
* Line 333: Lower case start of sentence
* Line 524: typo “is are”
* Line 534: Add whitespace after “Figure 3”
* Line 602: There seems to be a comment that hasn’t been removed prior to submission

---

> ### Author Rebuttal · Authors · 2023-08-28
>
> Thank you for your valuable and constructive feedback. As we say in our "Summary of Rebuttal" comment, the research design and the dataset collection are the main contributions of the paper. The choice to report only simple deep-learning baselines was deliberate so that we could focus on our unique contribution. We plan to explore directions in predictive language and behavioral modeling in subsequent projects, while also opening this dataset up for other researchers to use.
>
> 1.	The NA option was included in the response options provided to the participants so that we did not accidentally encode the absence/irrelevance of an appraisal to be the same as a low-rated appraisal.
>
> 2.	Selection of appraisal dimensions: The choice of these appraisal dimensions is supported by previous studies that show certain appraisals that are particularly relevant to the context of consumption (Demir et al., 2009; Desmet, 2008; Desmet & Hekkert, 2007). Nevertheless, we also added two questions that we feel might elicit other appraisals (see Table 8- a) How did you feel when you used the product/service?, and b) What about the product/service that made you feel this way?).
>
> 3.	A comprehensive set of emotion prediction experiments was beyond the scope of our research project because our main contribution was the research design and the dataset collection. Our choice of appraisal dimensions was based on prior work. Furthermore, we expect that pleasantness and goal congruence predictions are the most accurate likely due to their close alignment with positive/negative valence words.
>
> 4.	Regarding label distribution: Sure, we would be happy to add the distribution of class labels. They are already in the online repo at https://anonymous.4open.science/r/PEACE-Reviews-1DCF
>
> 5.	Regarding further analysis: as mentioned above, we wanted to focus our paper on our major contribution - the research design and data collection of reviews annotated by participants. We are working on subsequent projects for further analyses such as topic modeling and using validated lexicons (e.g. LIWC) to identify what words and tokens are indicative of emotions and appraisals so as to better understand the emotional experience of reviews.
>
> References
>
> Desmet, P., & Hekkert, P. (2007). Framework of product experience. International journal of design, 1(1), 57-66.
>
> Desmet, P. M. (2008). Product emotion. In Product experience(pp. 379-397). Elsevier.
>
> Demir, E., Desmet, P. M., & Hekkert, P. (2009). Appraisal patterns of emotions in human-product interaction. International journal of design, 3(2).

---

### Official Review · Reviewer_5opM · 2023-08-09

**Typos Grammar Style And Presentation Improvements:** 1. The writing of this paper could be…
**Soundness:** 3

**Excitement:**

4: Strong: This paper deepens the understanding of some phenomenon or lowers the barriers to an existing research direction.

**Missing References:**

“A meta-analytic review of the associations between cognitive appraisals and emotions in cognitive appraisal theory” (Yeo and Ong, 2023). The appraisal dimensions used in this study are *all* found in Yeo and Ong’s work, but not mentioned at all here.

**Paper Topic And Main Contributions:**

This paper revolves around emotion appraisals, i.e., how people evaluate their situations that results in specific emotional experiences. A set of 20 cognitive appraisal dimensions were examined. Using a pilot study, the authors were able to settle on a framework in the format of both the presence of emotion prompts and questionnaires to elicit high-quality first-person narratives that are rich in both emotional and cognitive appraisals. Using this framework, the authors then created the PEACE-Reviews dataset, which consists of annotated autobiographical writing where participants describe their appraisals and emotions during the usage of products or services of personal significance. Subsequent analyses and experiments were performed on the dataset.

The paper has the following main contributions: 1) This work identifies and proposes a useful framework for eliciting emotions together with the appraisals behind the emotions through writing prompts. 2) The authors propose the PEACE-Reviews dataset, which is the first to present emotion and cognition along multiple dimensions along with person-level trait and demographic information. 3) The authors present baseline models on identifying emotions and cognitive appraisals from autobiographical writing using language models.

**Questions For The Authors:**

1. Lines 67-81 and lines 490-492: Why do you include the decision-making and person-level traits data if it’s not discussed in this paper? Can you talk about the decision-making and person-level traits data that you collected in more detail?

2. On lines 314-316, you mentioned that participants might feel multiple emotions during a consumption episode. However, in both the main study and the baseline modeling, you focus on the dominant emotion only. Why don’t you take into account multiple emotions in your dataset?

3. The discussion of the analyses from the pilot study is unclear and ambiguous. Can you discuss the results using LIWC22 in Sec 5.3 in more detail? The results from Figure 2 are barely discussed. For the main study, the authors mentioned that they chose the [questionnaire format] as the writing prompts based on results from the pilot study. But in Figure 2, the [questionnaire format] only surpasses the [review format] in terms of the expressions of “cognition”. For the expressions of “affect”, I don’t see significant differences in distribution between the two formats. In fact, for expressions of psychological states such as “fulfilled”, “acquire”, and “want”, the [review format] actually often expresses much more psychological states than the [questionnaire format], as manifested in Figure 2. Can the authors provide more explanations why the [questionnaire format] is better than the [review format] based on the results in Figure 2?

4. Lines 407-413: Can you explain why there were no significant differences between conditions with or without emotion prompts? Because based on Figure 2, it looks like there is a large difference between the [Review] and [Emotion + Review] formats, especially in terms of the expressions of psychological states.

5. Section 7.3: Why don’t you include a simpler baseline such as the logistic regression model, so that the performance of the transformer models can be more clearly compared to?

6. Section 7.3: In addition, can you provide more explanations and details of the results of the emotion prediction experiments? For example, why is the performance of the models decent just because they are base models? What results should we expect to observe on more advanced models? Also, what is the accuracy and F1 score for each emotion? I am curious whether the models are more capable in predicting “joy” and “disappointment”, since they are the largest dominant positive and negative emotions felt on your domain (lines 518-520). Can you provide more details on the discussion of these?

7. Section 7: Can you explain the appraisal prediction task more clearly? It seems like you’re only trying to predict the appraisals “pleasantness”, “goal conduciveness”, and “expectedness” only, as shown in Table 4? Why do you pick these 3 dimensions and not include the other 17?

8. Section 7.3: Similar to the discussion of results for emotion prediction, a more detailed discussion of results for appraisal prediction is needed.

9. In conclusion you mentioned that you only included 4 appraisal questions (goal relevance, goal conducive-ness, expectedness, and pleasantness) to elicit text responses about appraisals. Did you mention that anywhere in the experiments? The design of the modelling is very unclear.

**Reasons To Accept:**

1. This work presents PEACE-Reviews, which offers review texts annotated with rich first-person emotional variables such as cognitive appraisal dimensions and emotional intensity. The first-person perspective autobiographical writing in the dataset fills the gap of prior work which focuses on emotions and appraisals from a 3rd-person perceived perspective.

2. Prior to collecting the PEACE-Reviews dataset, a pilot study was conducted to scrutinize which kinds of writing prompts elicit high-quality text of emotions and cognitive appraisals. After inspecting the results from the pilot study, the authors subsequently adopted the framework of  [Emotion Prompts] + [Questionnaire Format] to elicit first-person reviews from participants.

3. The data collection process is well-designed: 1) The authors included a screening question before collecting the main dataset that serves as a proxy for product/service that are personally important to the participants. This increases the likelihood of participants having emotion experiences when using the product/service. 2) Lines 451-461: both positive and negative experiences from the same participant are recorded. This better accounts for individual differences in the dataset. 3) There are 700 participants in total that contribute to the PEACE-Reviews dataset, which is a good indicator of the diversity involved in this dataset.

**Reasons To Reject:**

1. The authors failed to explain many critical aspects of the study. For example, where do the 20 appraisal dimensions come from? Are they extracted from prior studies or are they newly proposed in this specific work? The authors did not successfully convince me of the theoretical grounding for the appraisal dimensions they chose as they didn’t even discuss the origins of the 20 appraisal dimensions they included in this study.

2. The discussion of the analyses and experimental results from the study remains unclear and ambiguous. Please refer to the “Questions for the Authors” section for my questions.

3. The writing of this paper could be improved to a great extent, both in terms of the wording as well as the formulation of paragraphs. See the “Typos Grammar Style and Presentation Improvements” section for suggestions for improvements.

**Reproducibility:**

4: Could mostly reproduce the results, but there may be some variation because of sample variance or minor variations in their interpretation of the protocol or method.

**Reviewer Confidence:**

5: Positive that my evaluation is correct. I read the paper very carefully and I am very familiar with related work.

---

> ### Author Rebuttal · Authors · 2023-08-28
>
> We are grateful for your insightful feedback and would be happy to address your suggestions in the camera-ready version of the paper. As we say in our "Summary of Rebuttal" comment, the research design and the dataset collection are the main contributions of the paper. We plan to explore directions in predictive language and behavioral modeling in subsequent projects, while also opening this dataset up for other researchers to use. Many of your concerns have already been addressed in the paper and the appendix.
>
> 1. As mentioned in Table 4, lines 140-143, 166-169, and 604-606, we have clarified our focus on four appraisal dimensions for most of the analysis: goal relevance, goal conduciveness, pleasantness, and expectedness. In lines 655-657, we motivate our choice of appraisal dimensions based on the context of consumption; we say that “we only included four appraisal questions (goal relevance, goal conduciveness, expectedness, and pleasantness) to elicit text responses about appraisals. Nevertheless, we selected these four appraisals because they are most relevant to the context of consumption (Desmet, 2008) and we feel that introducing more appraisal questions for participants to complete could result in poor quality responses due to fatigue.” These appraisal dimensions are not new; instead, they are based on decades of literature on cognitive appraisal and consumption. In the camera-ready version, we will add more details and offer a lengthier motivation for our choices. Other papers that motivate and describe our choice of appraisal dimensions are Demir et al., 2009; Desmet, 2008; Desmet & Hekkert, 2007. We apologize for leaving out the meta-analysis the reviewer has provided and can include it in the camera-ready version. We regret the reviewer’s perceived brevity in our rationale.
>
> 2. Answers to the questions:
>
> -1. Given that EMNLP focuses on computational linguistics, we’ve not included the person-level variables in this version of the dataset. For the sake of transparency, we still mention that this was collected. We’re exploring these variables for a subsequent project on the cognitive antecedents and linguistic predictors of decision-making, targeting a psychological journal. Thank you for your interest!
>
> -2. As mentioned in lines 431-433 and line 514, we collected information about eight emotions from the respondent, while we did ask what the “dominant” emotion was. This information is used to collect information regarding cognitive appraisal, thereby allowing us to relate the cognitive antecedents of the dominant emotion to their consequence. We could not have asked these questions for each of the emotions the respondent experienced to be mindful of respondent fatigue.
>
> -3. Thank you for your keen observation and great question! These would be important directions to investigate in a follow-up study, as they reflect findings from preliminary data, based on 18 responses. But in general, data inspection revealed that in the review format, the emotion scores may reflect authors’ subjective assessment of the product, rather than their own emotional experience. For instance “bad food” is a different, and less useful text for the study of appraisal, as compared to “horribly upset.” We decided to focus our paper on the dataset collection and save these interesting explorations for future work.
>
> -4. For lines 407-413, we were referring to the comparison between [Questionnaire] and [Emotion + Questionnaire] format. We have conducted Wilcoxon signed-rank tests and found that there were no significant differences across LIWC categories except for ‘cognition’ in which the [Emotion + Questionnaire] format yielded greater cognition words. The reason would be because of the high variance in the measurements (based on 18 reviews). We will state more clearly in the camera-ready version of the paper.
>
> -5. Thank you for raising this suggestion. We’d like to highlight that while our main contribution is the research design and dataset collection, we went the extra mile to add more sophisticated deep-learning models. We hope the reviewers are satisfied with our efforts.
>
> -6. Thank you for your suggestion. As we noted on line 599, model performances on our dataset are consistent. A comprehensive set of emotion prediction experiments was beyond the scope of our research project because our main contribution is the research design and the dataset collection. Our emotion prediction experiments aim to establish baseline evaluations with state-of-the-art models on this novel dataset. Given its unique emotion classes, direct comparisons with other datasets are challenging. However, a close comparison is the ISEAR dataset (Scherer, & Wallbott, 1994), where XLNet achieved a 73.0% accuracy in seven-way emotion classification (Adoma et al., 2020). Notably, our dataset has imbalanced emotion classes, with joy and disappointment dominating. Using XLNet, we recorded accuracies of 95.7% and 92.0% for these emotions, respectively. As anticipated, the models excel in predicting these emotions but occasionally misclassify them into other positive or negative categories. This suggests models capture valence and pleasantness but struggle to differentiate specific emotions within a valence. We would be happy to clarify this in the camera-ready version.
>
> -7. Answers 7 and 9: We appreciate the feedback. We focused on four appraisal questions: 1. goal relevance, 2. goal conduciveness, 3. expectedness, and 4. pleasantness, backed by prior research (Demir et al., 2009; Desmet, 2008; Desmet & Hekkert, 2007). We also gathered responses to two other questions (see Table 8) and ratings for all 20 appraisal dimensions. We apologize for leaving out the meta-analysis the reviewer has provided and can include it in the camera-ready version. For classification, we prioritized dimensions with text responses. In the update, we'll expand the classification tasks and provide a distribution of class labels.
>
> -8. Answer 8: Our appraisal prediction experiments aim to establish baselines using top-tier models. Our choice of appraisal dimensions was based on prior work as cited above. We sincerely apologize for missing the reference of the meta-analysis the reviewer has provided and will include it in the camera-ready version. Furthermore, pleasantness and goal congruence predictions are the most accurate, likely due to their close alignment with positive/negative valence words.
>
> Finally, we sincerely apologize for the typos that remain in the paper and would be glad to address these in the camera-ready version. Thank you for your help in helping us to clarify the impact of our paper.
>
> References
>
> Adoma, A. F., Henry, N. M., & Chen, W. (2020, December). Comparative analyses of bert, roberta, distilbert, and xlnet for text-based emotion recognition. In 2020 17th International Computer Conference on Wavelet Active Media Technology and Information Processing (ICCWAMTIP) (pp. 117-121). IEEE.
>
> Desmet, P., & Hekkert, P. (2007). Framework of product experience. International journal of design, 1(1), 57-66.
>
> Desmet, P. M. (2008). Product emotion. In Product experience(pp. 379-397). Elsevier.
>
> Demir, E., Desmet, P. M., & Hekkert, P. (2009). Appraisal patterns of emotions in human-product interaction. International journal of design, 3(2).
>
> Scherer, K. R., & Wallbott, H. G. (1994). Evidence for universality and cultural variation of differential emotion response patterning. Journal of personality and social psychology, 66(2), 310.

---

### Meta-Review · Area_Chair_3zrh · 2023-09-18

**Recommendation:** 3

**Metareview:**

Thank you to authors for their work and reviewers for their reviews. This paper builds a new data set in which participants describe their appraisals and emotions during the usage of products or services of personal significance.

Pros:
- Reviewers agree that the data set, which aims to capture self-reported cognitive appraisals and emotions is novel in design and captures dimensions not present in current data sets
- Reviewers agree that the data set collection is well-thought-out, with a pilot study to determine the best format and contextualization within prior work

Cons:
- After reviewing the paper myself as well as the reviews and the author responses, I am missing some of the main motivation of the work. The authors state: "Therefore, the lack of a product/service review dataset containing emotionally-rich variables such as cognitive appraisals and emotional intensity is a primary motivation for the proposed dataset." While reviewers (and I) agree that a dataset like this does not currently exist, that does not address why such a data set should exist. The only direct use case I can think of for studying emotions in product reviewers is to improve targeted advertising. If the authors have other use cases in mind, I would suggest adding them to the paper if it is accepted.
- Emotion classification in general is an ethically complex topic: [1] outlines some of the debates regarding NLP specifically. I think the paper can more thoroughly address the ethical implications of their work, not only in the ethical implications section, but also in clarifying the limitations of their dataset throughout the work, for example, how post-hoc descriptions of emotions may differ from actual emotions. Given that the authors are planning to release the data, it would be useful to add more guidance on how future researchers should and should not use it to help prevent future researchers from making misinterpretations or inaccurate claims based on this data.
- Reviewers note areas for clarification in the details of the data set. These look like minor clarifications to me, some of which are addressed in the appendix and could be moved to the main paper for a final version.

Overall, I think this is a very interesting and well-collected data set and my concerns with the work are primarily about potential misuse.

[1] Mohammad, Saif M. "Ethics sheet for automatic emotion recognition and sentiment analysis." Computational Linguistics 48.2 (2022): 239-278

---

### Decision · Program_Chairs · 2023-10-07

**Decision:**

Accept-Findings

**Comment:**

Thank you to authors for their work and reviewers for their reviews. This paper builds a new data set in which participants describe their appraisals and emotions during the usage of products or services of personal significance.

Pros:
- Reviewers agree that the data set, which aims to capture self-reported cognitive appraisals and emotions is novel in design and captures dimensions not present in current data sets
- Reviewers agree that the data set collection is well-thought-out, with a pilot study to determine the best format and contextualization within prior work

Cons:
- After reviewing the paper myself as well as the reviews and the author responses, I am missing some of the main motivation of the work. The authors state: "Therefore, the lack of a product/service review dataset containing emotionally-rich variables such as cognitive appraisals and emotional intensity is a primary motivation for the proposed dataset." While reviewers (and I) agree that a dataset like this does not currently exist, that does not address why such a data set should exist. The only direct use case I can think of for studying emotions in product reviewers is to improve targeted advertising. If the authors have other use cases in mind, I would suggest adding them to the paper if it is accepted.
- Emotion classification in general is an ethically complex topic: [1] outlines some of the debates regarding NLP specifically. I think the paper can more thoroughly address the ethical implications of their work, not only in the ethical implications section, but also in clarifying the limitations of their dataset throughout the work, for example, how post-hoc descriptions of emotions may differ from actual emotions. Given that the authors are planning to release the data, it would be useful to add more guidance on how future researchers should and should not use it to help prevent future researchers from making misinterpretations or inaccurate claims based on this data.
- Reviewers note areas for clarification in the details of the data set. These look like minor clarifications to me, some of which are addressed in the appendix and could be moved to the main paper for a final version.

Overall, I think this is a very interesting and well-collected data set and my concerns with the work are primarily about potential misuse.

[1] Mohammad, Saif M. "Ethics sheet for automatic emotion recognition and sentiment analysis." Computational Linguistics 48.2 (2022): 239-278